# An IL-2-grafted antibody immunotherapy with potent efficacy against metastatic cancer

Dilara Sahin[1], Natalia Arenas-Ramirez[1], Matthias Rath[1], Ufuk Karakus[1], Monika Hümbelin[1], Merel van Gogh[2], Lubor Borsig[2] & Onur Boyman [1,3✉]

Modified interleukin-2 (IL-2) formulations are being tested in cancer patients. However, IL-2 immunotherapy damages IL-2 receptor (IL-2R)-positive endothelial cells and stimulates IL-2Rα (CD25)-expressing lymphocytes that curtail anti-tumor responses. A first generation of IL-2Rβ (CD122)-biased IL-2s addressed some of these drawbacks. Here, we present a second-generation CD122-biased IL-2, developed by splitting and permanently grafting unmutated human IL-2 (hIL-2) to its antigen-binding groove on the anti-hIL-2 monoclonal antibody NARA1, thereby generating NARA1leukin. In comparison to hIL-2/NARA1 complexes, NARA1leukin shows a longer in vivo half-life, completely avoids association with CD25, and more potently stimulates CD8$^+$ T and natural killer cells. These effects result in strong anti-tumor responses in various pre-clinical cancer models, whereby NARA1leukin consistently surpasses the efficacy of hIL-2/NARA1 complexes in controlling metastatic disease. Collectively, NARA1leukin is a CD122-biased single-molecule construct based on unmutated hIL-2 with potent efficacy against advanced malignancies.

[1] Department of Immunology, University Hospital Zurich, CH-8091 Zurich, Switzerland. [2] Institute of Physiology, University of Zurich, CH-8057 Zurich, Switzerland. [3] Faculty of Medicine, University of Zurich, CH-8006 Zurich, Switzerland. ✉email: onur.boyman@uzh.ch

Owing to its ability to stimulate effector-type immune cells, high-dose interleukin-2 (IL-2) was the first approved immunotherapy for metastatic cancer. However, IL-2 immunotherapy is limited by dose-dependent adverse events, which are particularly evident and severe at the high doses necessary to achieve clinical efficacy[1]. These adverse effects in different organs are due to endothelial cell damage, also termed vascular leak syndrome. Moreover, IL-2 stimulates suppressor-type immune cells, which in turn curtail anti-tumor effector cells[2–4]. Lastly, due to its low molecular weight of 15.5 kDa, recombinant IL-2 exhibits a short in vivo half-life[1]. Insight into the biology of IL-2 and the IL-2 receptor (IL-2R) has provided possibilities to dissect the beneficial from the unwanted effects of IL-2.

IL-2 can stimulate several leukocyte subsets, including suppressor-type CD4[+] forkhead box p3 (Foxp3)[+] regulatory T (Treg) cells, cytotoxic CD8[+] T cells, and natural killer (NK) cells[5,6]. IL-2 exerts its effects through signaling via two different IL-2Rs. A dimeric IL-2R formed by association of the subunits IL-2Rβ (CD122) and IL-2Rγ (CD132; also termed common γ chain, $\gamma_c$) is mainly expressed on antigen-experienced (memory) CD8[+] T and NK cells. The addition of a third subunit, named IL-2Rα (CD25), results in the trimeric IL-2R, which is expressed highly on CD4[+] CD25[+] Foxp3[+] Treg cells[1,7]. Upon binding of IL-2, signal transduction relies on CD122 and $\gamma_c$, whereas CD25 is dispensable for signaling but instead serves to increase binding affinity for IL-2.

Association of IL-2 with a specific anti-IL-2 monoclonal antibody (mAb), thus forming IL-2/anti-IL-2 mAb complexes (IL-2cx), can overcome the aforementioned shortcomings of high-dose IL-2[8–11]. To proceed toward clinical development, we recently generated and humanized NARA1, a mAb specifically associating with the CD25-binding site of human IL-2 (hIL-2), thus acting as a "CD25 mimobody" and forming CD122-biased hIL-2/NARA1 complexes[12]. Compared to unbound hIL-2, such hIL-2/NARA1 complexes prolong the half-life of hIL-2 and preferentially activate CD122[high] CD8[+] T and NK cells, whereas

association of hIL-2 with CD25-expressing Treg and endothelial cells is hampered as long as hIL-2 is bound to NARA1. Overall, these effects result in superior CD8[+] T cell-mediated tumor control in several melanoma mouse models[12].

To form hIL-2/NARA1 complexes, hIL-2 and NARA1 are premixed prior to injection and associate with high affinity (dissociation constant $K_D = 1.4 \times 10^{-9}$ M). Although the complexes can be detected in the serum for 24–48 hours, the two components can dissociate resulting in unbound hIL-2 and NARA1, which undermines the advantages of hIL-2/NARA1 complexes. Permanent coupling of hIL-2 to NARA1 would prevent in vivo dissociation.

Based on the crystal structure of the hIL-2/NARA1 complex[12], we previously designed, produced, and characterized several immunocytokine fusion proteins (FPs) where flexible glycine-serine (GS) linkers connected hIL-2 to the variable region of the light or heavy chain of NARA1[13]. However, such hIL-2–NARA1 FPs proved challenging to produce and were inferior to hIL-2/NARA1 complexes in vivo. Thus, we adopted a different strategy, integrating hIL-2 directly into its antigen-binding groove on NARA1[13]: We grafted hIL-2 to the complementarity-determining region 1 of the light chain (L-CDR1) of NARA1, which resulted in a single and stable molecule termed NARA1leukin.

In this work, we show that NARA1leukin completely abolishes the binding of hIL-2 to CD25 in vitro and in vivo. Furthermore, NARA1leukin increases the in vivo half-life of IL-2 and redirects IL-2's stimulatory activity to CD122[high] effector cells translating to potent anti-cancer responses in several preclinical tumor models.

## Results

**Development of NARA1leukin.** Based on the 3D-crystal structure of the hIL-2/NARA1 complex we determined an interaction site, the B-C loop of hIL-2 and L-CDR1 of humanized NARA1 (hNARA1), which was suitable as an anchoring point. We dissociated hIL-2 between lysine 76 (K76) and asparagine 77 (N77)

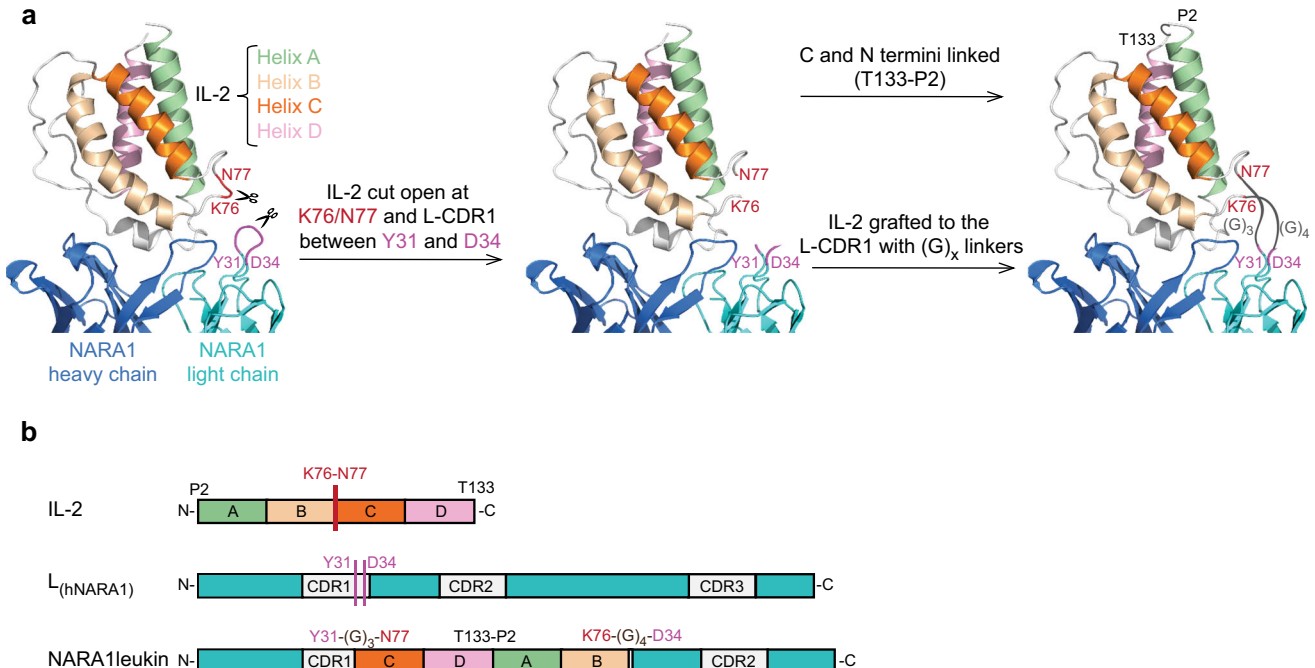

**Fig. 1 Design of NARA1leukin. a** Graphical representation of strategy followed for generation of NARA1leukin. hIL-2: helix A, green; helix B, yellow; helix C, dark orange; helix D, pink; and NARA1: heavy chain, blue; light chain, cyan. **b** Schematic representation of strategy followed for generation of NARA1leukin showing the engineered protein (color code as in **a**).

as well as hNARA1's L-CDR1 between tyrosine 31 (Y31) and aspartic acid 34 (D34). Subsequently, we inserted hIL-2 in the frame of L-CDR1, bridging hNARA1's Y31 and D34 to hIL-2's N77 and K76 residues, respectively (Fig. 1a). To conserve the non-covalent interactions between hIL-2 and hNARA1, we replaced the amino acids at positions 32 and 33 of L-CDR1 by short glycine linkers, $(G)_4$ and $(G)_3$. Moreover, we linked proline 2 (P2) and threonine 133 (T133) at the N- and C-termini of hIL-2, thereby creating a continuous construct (Fig. 1b). We termed the resultant molecule NARA1leukin. NARA1leukin[high] was a stable molecule with minimal aggregate formation upon purification, which greatly facilitated production.

**NARA1leukin avoids CD25-binding and improves CD122 selectivity in vitro.** To assess whether the functional integrity of hIL-2 was preserved upon engraftment onto hNARA1, we tested NARA1leukin's ability to associate with defined anti-hIL-2 mAbs and recombinant hCD25. To this end, we devised a sandwich enzyme-linked immunosorbent assay (ELISA) with the anti-hIL-2 mAbs clone 5344 as capture mAb (binding hIL-2's CD122/$\gamma_c$ epitope)[10,12] and MAB202 as detection mAb (associating with a hIL-2 region distinct from its IL-2R-binding epitopes); in this ELISA, NARA1leukin bound with similar affinity as free hIL-2 and hIL-2/NARA1 complexes (Fig. 2a). Contrarily, in a sandwich ELISA with plate-bound recombinant hCD25 and MAB202 as detection mAb, NARA1leukin did not give a signal, even at high concentrations (Fig. 2b), indicating abolishment of hCD25 binding.

Next, we assessed the ability of NARA1leukin to associate with and signal via dimeric CD122–$\gamma_c$ and trimeric CD25–CD122–$\gamma_c$ IL-2Rs. Upon stimulation with NARA1leukin, human CD8[+] T cells and mouse CD8[+] T and NK cells showed similar levels of phosphorylated STAT5 (pSTAT5) compared to free hIL-2 and hIL-2/NARA1 complexes, while a 30-fold superior stimulation was noted for human NK cells (Fig. 2c, d). Conversely, the stimulation of CD4[+] CD25[+] Treg cells was decreased by 30-fold for human and by 8000-fold for mouse Treg cells when stimulated with NARA1leukin compared to hIL-2/NARA1 complexes (Fig. 2c, d), due to lower expression of dimeric CD122–$\gamma_c$ IL-2Rs on Treg cells. Collectively, these data demonstrate that NARA1leukin exhibits improved in vitro CD122 bias and CD25 avoidance, when compared to hIL-2/NARA1 complexes.

**NARA1leukin shows prolonged bioactivity and increased CD122 bias in vivo.** Upon a single injection of NARA1leukin to mice, we observed detectable hIL-2 serum levels up to 72-96 hours, compared to 24–48 hours for hIL-2/NARA1 complexes (Fig. 3a). In vivo dose escalation studies demonstrated a higher potency of NARA1leukin in stimulating CD122[high] cells. The increase in cell counts and proliferation (as measured by Ki67 positivity) observed with hIL-2/NARA1 complexes was readily achieved by 4-fold lower doses of hIL-2 in NARA1leukin. The optimal dose of NARA1leukin was determined to correspond to 0.5 µg hIL-2 equivalent, as higher doses did not further increase CD8[+] T and NK cell expansion but showed a trend toward toxicity (Supplementary Fig. 1).

To further characterize the in vivo effects of NARA1leukin, we performed a pharmacodynamic study where pSTAT5, proliferation, and cell counts were analyzed after a single injection of hIL-2, hIL-2/NARA1 complexes, and NARA1leukin (Fig. 3b). In mice receiving NARA1leukin, pSTAT5 signals in CD8[+] T and NK cells persisted steadily for 2 days, unlike hIL-2 and hIL-2/NARA1 complexes where pSTAT5 signals were significantly reduced already on day 1 (Fig. 3c, d). This activation subsequently translated to significant expansion of CD8[+] T and NK cells, which correlated with high Ki67 levels indicating vigorous proliferation (Fig. 3c, d). Contrarily, NARA1leukin did not induce any pSTAT5 or Ki67 upregulation in CD4[+] CD25[+] Treg cells (Fig. 3e), verifying its improved selectivity in vivo.

**NARA1leukin improves anti-tumor immune responses.** Based on our aforementioned results, we hypothesized that NARA1leukin should exert anti-tumor effects at lower doses than hIL-2/NARA1 complexes. In the intradermal B16-F10 melanoma model, we compared the optimal dose of NARA1leukin (0.5 µg hIL-2 equivalent) with an equivalent dose (0.5 µg/2.5 µg) and a three-fold higher dose (1.5 µg/15 µg, corresponding to the optimal dose[12]) of hIL-2/NARA1 complexes (Fig. 4a). In this model, NARA1leukin required less frequent injections and achieved comparable tumor control at 27% of the total dose of hIL-2/NARA1 complexes (Fig. 4b, c). To assess the mechanism of NARA1leukin's anti-tumor effects, we analyzed the composition of tumor-infiltrating lymphocytes (TILs) with a particular interest in the ratio of CD8[+] T to CD4[+] Foxp3[+] Treg cells. Total counts of CD44[+] CD8[+] T and NK cells in tumors, tumor-draining lymph nodes (TDLNs), and spleens were similarly increased in hIL-2/NARA1 complex- and NARA1leukin-treated animals (Fig. 4d and Supplementary Fig. 2). However, counts of CD4[+] CD25[+] Foxp3[+] Treg cells were considerably lower in tumors, TDLNs, and spleens of mice receiving NARA1leukin compared to hIL-2/NARA1 complexes (Fig. 4d). These changes resulted in a favorable ratio of CD44[+] CD8[+] T to Treg cells for NARA1leukin (Fig. 4e). Although low in numbers, Foxp3[+] Treg cells remained functional in NARA1leukin-treated animals; thus, the depletion of Treg cells in Foxp3[DTR] mice by using diphtheria toxin synergized with NARA1leukin therapy to further improve tumor control (Supplementary Fig. 2).

In animals receiving IL-2 immunotherapy, the majority of intratumoral CD8[+] T cells were CD44[+] and expressed high levels of homing molecules, such as CD62L and CXCR3 (Supplementary Fig. 2). Tumor-infiltrating CD8[+] T cells also showed decreased abundance of TOX, programmed cell death protein-1 (PD-1), T cell immunoglobulin, mucin domain-3 (TIM-3), and killer cell lectin-like receptor subfamily G member 1 (KLRG1), indicating the cells remained functional effector cells (Fig. 5f–i and Supplementary Fig. 2). These effects were observed with hIL-2/NARA1 complexes and were even more pronounced with NARA1leukin.

We did not observe significant IL-2-mediated toxicity in mice receiving hIL-2/NARA1 complexes or NARA1leukin. Thus, pulmonary wet weight, as well as serum levels of aspartate aminotransferase (AST) and alanine aminotransferase (ALT), remained low and comparable to mice treated with hIL-2/NARA1 complexes (Supplementary Fig. 2).

To compare NARA1leukin with another IL-2-based one-molecule therapeutic approach, we used H9, which is a CD122-biased IL-2 mutein[14], covalently linked to human Fc (termed H9-Fc). H9-Fc stimulated CD8[+] T cells in vitro to a similar extent as NARA1leukin and slightly better than hIL-2/NARA1 complexes; conversely, both H9-Fc and NARA1leukin were better than hIL-2 and hIL-2/NARA1 complexes at avoiding the activation of CD4[+] CD25[+] T cells, with NARA1leukin being even more selective than H9-Fc (Supplementary Fig. 3). However, the stimulation of effector cells in vivo was less pronounced following H9-Fc compared to hIL-2/NARA1 complexes and NARA1leukin (Supplementary Fig. 3). In the intradermal B16-F10 melanoma model, H9-Fc led to a modest reduction of tumor growth, which was surpassed by hIL-2/NARA1 complexes and NARA1leukin (Supplementary Fig. 3).

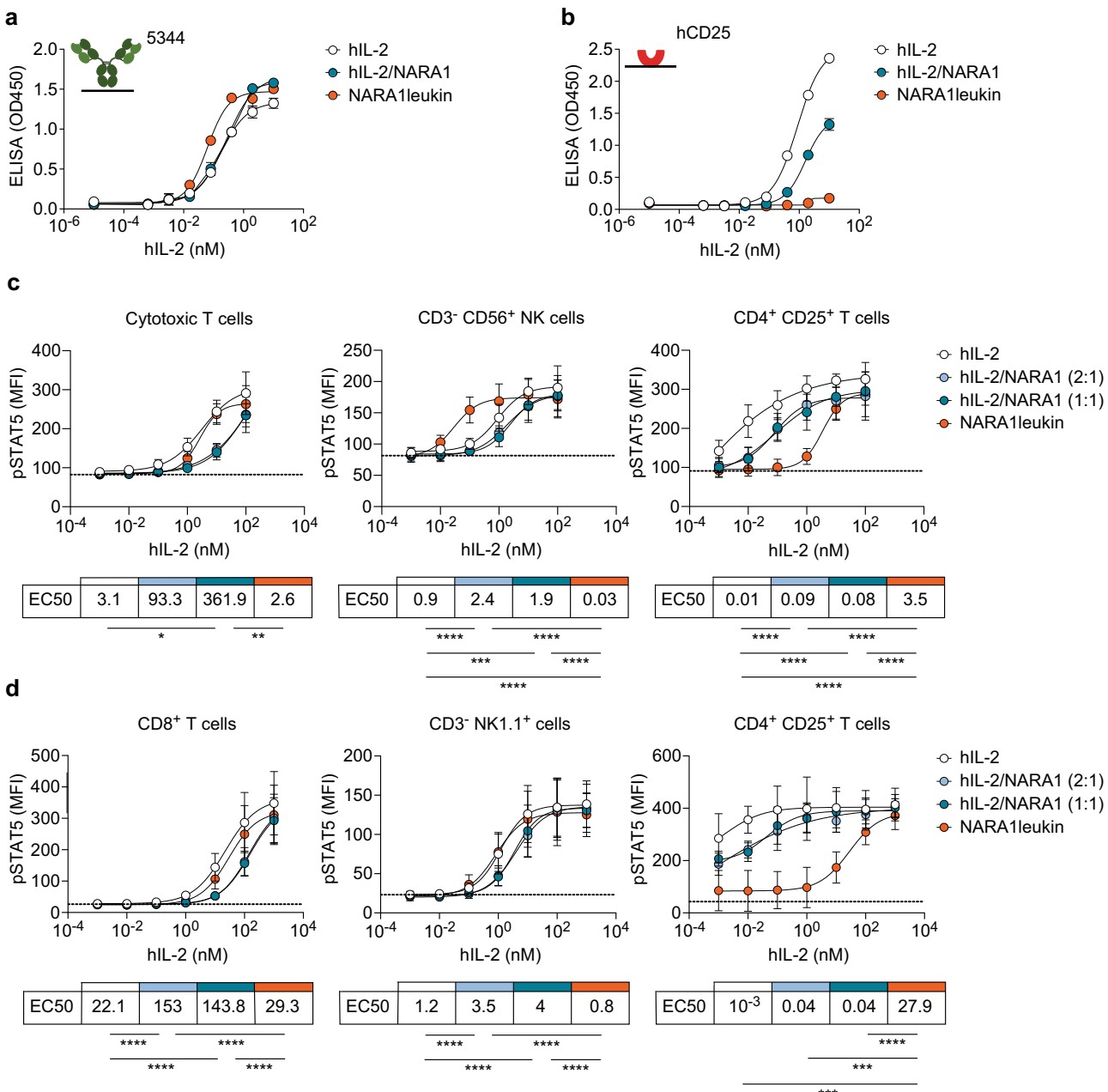

**Fig. 2 In vitro characteristics of NARA1leukin. a, b** Sandwich ELISA coating anti-hIL-2 mAb 5344 (**a**) or hCD25 (**b**) and detecting with anti-hIL-2 mAb MAB202 shows binding of titrated hIL-2, hIL-2/NARA1 complexes, and NARA1leukin. **c, d** Phosphorylated STAT5 (pSTAT5) levels of human (**c**) and mouse (**d**) immune cell subsets responding to titrated hIL-2, hIL-2/NARA1 complexes, and NARA1leukin. 1:1 and 2:1 indicate molar ratios of cytokine to antibody of hIL-2/NARA1 complexes. Dotted lines indicate mean fluorescent intensity (MFI) of pSTAT5 of each cell subset following incubation in media for 15 min. Half maximal effective concentration (EC50) of each stimulant is presented in the table below. Differences between curves were analyzed using one-way ANOVA followed by Tukey's multiple comparison test (**c, d**). (**c**) $*p = 0.0112$, $**p = 0.0082$, $****p < 0.0001$, $***p = 0.0002$. (**d**) $****p < 0.0001$, $***p = 0.0001$ (hIL-2), $***p = 0.0006$ (hIL-2/NARA1). Data are presented as mean ± SD of two (**a, b**), four (**c**), or three (**d**) independent experiments. Source data are provided as a Source Data file.

**NARA1leukin immunotherapy reduces metastatic colonization by circulating tumor cells.** As immunotherapies are mainly used for metastatic malignancies, we tested the therapeutic efficacy of NARA1leukin in a pulmonary melanoma model where B16-F10 cells were injected intravenously (Fig. 5a). Interestingly, pulmonary B16-F10 nodules were better controlled by NARA1leukin treatment, with an overall reduction of pulmonary nodules by more than 84% compared to PBS, whereas high-dose hIL-2/NARA1 complexes only achieved 51% reduction (Fig. 5b–d). This effect led to complete abolishment of lung nodules in 10% of mice

treated with NARA1leukin, whereas high-dose hIL-2/NARA1 complexes did not result in complete clearance pulmonary nodules. Analysis of immune cells present in pulmonary B16-F10 nodules showed a preferential increase in CD44+ CD8+ T and NK cells over CD25+ Foxp3+ CD4+ Treg cells (Supplementary Fig. 4), which was similar to the composition we observed for cutaneous B16-F10 nodules.

Next, we established a more physiologic metastasis model where lymph node metastasis is observed in 75% of the untreated mice after surgical removal of the primary B16-F10 tumor from

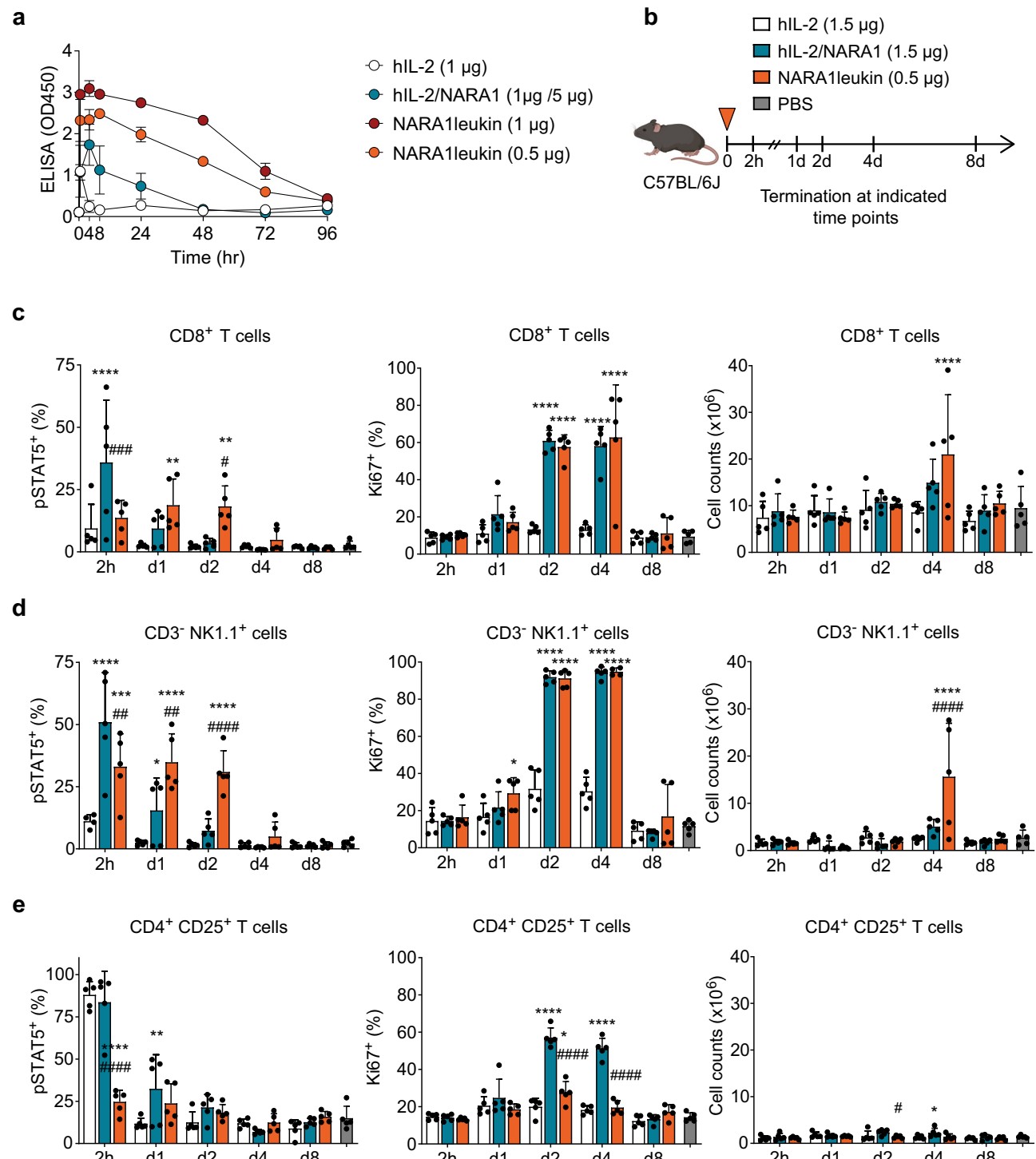

**Fig. 3 In vivo characteristics of NARA1leukin. a** Mice received a single injection of hIL-2 (1 μg), hIL-2/NARA1 complexes (1 μg/5 μg), or NARA1leukin (0.5 μg or 1 μg hIL-2 equivalent). Blood samples were collected at the indicated time points and the presence of hIL-2 in serum was assessed using a sandwich ELISA coating anti-hIL-2 mAb 5344 and detecting with anti-hIL-2 mAb MAB202. Data are represented as mean ± SD of two independent experiments, $n = 2–6$ mice/group. **b** Experimental scheme. Mice received a single injection (arrowhead) of hIL-2 (1.5 μg), hIL-2/NARA1 complexes (1.5 μg/15 μg), or NARA1leukin (0.5 μg hIL-2 equivalent) and were assessed at indicated time points after the injection (h: hours, d: days). **c–e** Spleens were analyzed by flow cytometry to determine pSTAT5, Ki67, and total cell counts of CD8+ T cells (**c**), CD3- NK1.1+ NK cells (**d**), and CD4+ CD25+ T cells (**e**). Differences between groups at the same time point were analyzed using two-way ANOVA followed by Tukey's multiple comparison test (**c–e**). The comparisons of hIL-2/NARA1 and NARA1leukin to hIL-2 are indicated by asterisks (*), the differences between hIL-2/NARA1 and NARA1leukin are indicated by hashtags (#) on top of the columns; non-significant p-values are not indicated. (**c**) ****$p < 0.0001$, ###$p = 0.0002$, **$p = 0.0082$ (d1) **$p = 0.0085$ (d2) #$p = 0.0168$. (**d**) ****$p < 0.0001$ ***$p = 0.0006$, ##$p = 0.0032$ (2 h), *$p = 0.0412$ (pSTAT5), ##$p = 0.013$ (d1), ####$p < 0.0001$, *$p = 0.0252$ (Ki67) (**e**) ****$p < 0.0001$, ####$p < 0.0001$, **$p = 0.0016$, *$p = 0.0227$ (d2), #$p = 0.0302$, *$p = 0.0337$ (d4). Data are presented as mean ± SD of three independent experiments, $n = 5$ mice/group. Source data are provided as a Source Data file.

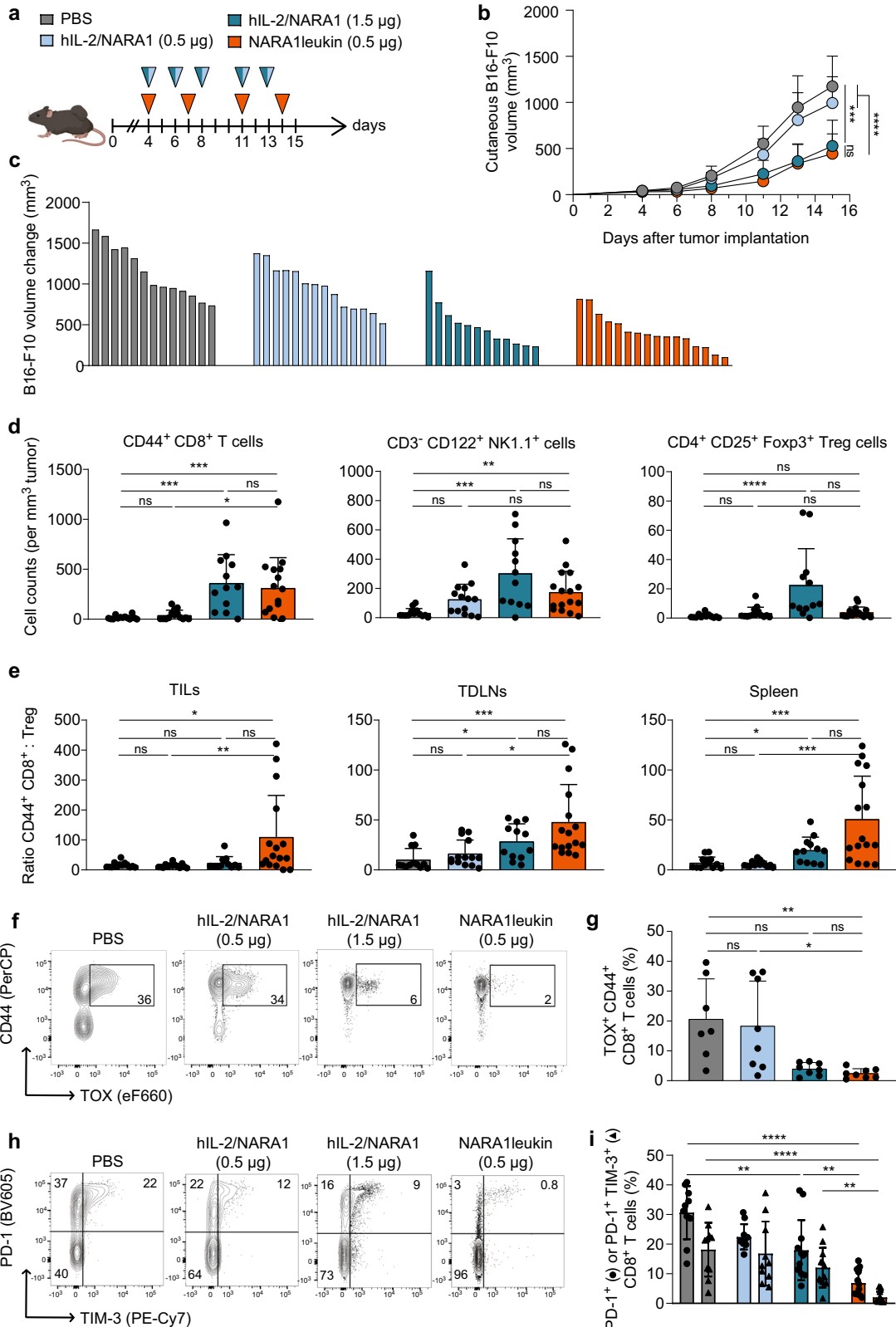

the skin (Fig. 5e). Neoadjuvant treatment with NARA1leukin prior to surgery abolished metastasis formation in 93% of the mice, generating the highest survival rate among the different treatment groups (Fig. 5f).

Furthermore, we assessed the synergistic potential of NAR-A1leukin with peptide vaccination in a B16-F10 model allowing observation of both primary and metastatic lesions (Fig. 5g).

Although control of primary tumors was comparable between high-dose hIL-2/NARA1 complexes and NARA1leukin, lung nodules were more strongly inhibited by NARA1leukin (Fig. 5h–j). The combination with peptide vaccination improved the control of primary tumors with both IL-2 immunotherapies, with NARA1leukin showing some mild benefit over hIL-2/NARA1 complexes (Fig. 5h). Furthermore, the lung nodules were

**Fig. 4 Anti-tumor immune responses generated by NARA1leukin. a** Treatment scheme. Mice were injected with $10^6$ B16-F10 melanoma cells intradermally and treated with PBS, hIL-2/NARA1 complexes (0.5 μg/2.5 μg or 1.5 μg/15 μg, three times a week), or NARA1leukin (0.5 μg hIL-2 equivalent, two times a week) from day 4 to 14. Spleens, tumor-draining lymph nodes (TDLNs), and tumor-infiltrating lymphocytes (TILs) were analyzed at day 15 by flow cytometry for the indicated immune cell subsets. **b** Tumor growth curves of intradermal B16-F10 melanoma until day 15. ****$p < 0.0001$, ***$p = 0.0001$. **c** Change in tumor volumes in individual mice between days 4 and 15. **d** Cells counts of CD44$^+$ CD8$^+$ T, CD3$^-$ CD122$^+$ NK1.1$^+$ NK, and CD4$^+$ CD25$^+$ Foxp3$^+$ Treg cells per mm$^3$ of tumor are indicated. CD8: ***$p = 0.001$ (NARA1leukin), ***$p = 0.0005$ (hIL-2/NARA1), *$p = 0.0166$; NK: **$p = 0.0069$, ***$p = 0.0003$; Treg: ****$p < 0.0001$. **e** Ratios of CD44$^+$ CD8$^+$ T to CD4$^+$ CD25$^+$ Foxp3$^+$ Treg cells for indicated organs are shown. TILs: *$p = 0.0333$, **$p = 0.0043$; TDLNs: ***$p = 0.0002$, *$p = 0.0238$ (PBS), *$p = 0.0166$ (hIL-2/NARA1); Spleen: ***$p = 0.0004$ (PBS), *$p = 0.0389$, ***$p = 0.0001$ (hIL-2/NARA1). **f-i** Intratumoral CD8$^+$ T cells were analyzed by flow cytometry for CD44 versus TOX (**f, g**) and PD-1 versus TIM-3 (**h, i**). Shown are representative flow cytometry plots (**f, h**) and percentages (**g, i**) of CD8$^+$ T cells. **g** **$p = 0.0049$, *$p = 0.0171$. **i** ****$p < 0.0001$, **$p = 0.0011$ (PBS), **$p = 0.003$ (●), **$p = 0.0089$ (▲). Data are presented as mean ± SD of four (**b–e, h** and **i**) or three (**f, g**) independent experiments,) $n = 15$ (PBS and hIL-2/NARA1 0.5 μg), $n = 12$ (hIL-2/NARA1 1.5 μg), $n = 16$ (NARA1leukin) (**b**), $n = 13$ (PBS), $n = 14$ (hIL-2/NARA1 0.5 μg), $n = 12$ (hIL-2/NARA1 1.5 μg), $n = 16$ (NARA1leukin) (**d, e**), $n = 7$ (PBS), $n = 8$ (hIL-2/NARA1 0.5 μg/1.5 μg and NARA1leukin) (**g**), $n = 10$ (PBS and hIL-2/NARA1 0.5 μg), $n = 12$ (hIL-2/NARA1 1.5 μg and NARA1leukin) (**i**) mice/group. Differences between groups at the same time point were analyzed using mixed-effects analysis with Greenhouse–Geisser correction followed by Tukey's multiple comparison test (**b**). Differences in immune cell subsets between groups were analyzed using Kruskal–Wallis test followed by Dunn's multiple comparison test (**d, e**, and **g**) or by two-way ANOVA followed by Tukey's multiple comparison test (**i**). ns, not significant. Source data are provided as a Source Data file.

robustly inhibited by NARA1leukin and vaccine combination, with more than 50% of treated mice showing no evidence of pulmonary nodules at the endpoint (Fig. 5i, j). Notably, NARA1leukin monotherapy was able to achieve control of pulmonary lesions to a comparable extent as combined high-dose hIL-2/NARA1 complexes with vaccination (Fig. 5i, j).

**Efficacy of NARA1leukin against spontaneous metastasis.** Our observations in induced metastatic models led us to direct our investigations to models that show natural metastatic spreading of the primary tumors. We assessed the efficacy of NARA1leukin in the poorly immunogenic Lewis lung carcinoma (LLC) and 4T1 mammary cancer models where spontaneous lung metastases form following injection of a primary tumor to the skin or mammary fat pad, respectively. In the LLC model, the progression of both primary and metastatic lesions were inhibited by NARA1leukin, with a most impressive effect on lung metastases (Fig. 6a–f). Thus, 92% of NARA1leukin-treated mice were clear of metastasis at the endpoint compared to only 8% and 36% for untreated and hIL-2/NARA1 complex-treated mice, respectively (Fig. 6d–f).

In the 4T1 tumor model, both IL-2 immunotherapies only mildly affected the growth of the primary tumor (Fig. 6g–i). However, NARA1leukin completely inhibited metastasis formation in almost half of treated mice, whereas the metastatic burden was only decreased, but not cleared, by treatment with hIL-2/NARA1 complexes (Fig. 6j–l).

## Discussion

To overcome the limitations of high-dose hIL-2 immunotherapy for metastatic cancer, different avenues have been pursued in the past years, including the development of muteins of IL-2 or its receptors as well as association of IL-2 to large molecules[14–21]. These approaches aim to increase anti-tumor efficacy and reduce IL-2-associated adverse effects by directed IL-2R binding, or targeting to the tumor site, and some of these molecules are currently in clinical trials[17,19–21]. Several muteins of hIL-2 have been generated to ablate the interaction of hIL-2 with CD25. However, such approaches suffer from reduced stability and/or short half-life, requiring daily injections or conjugation to a large molecule, such as an antibody[14,22]. Moreover, introduced mutations create the risk of immunogenicity and generation of neutralizing anti-drug antibodies (ADAs), which limit the clinical use of hIL-2 muteins. The hIL-2 mutein BAY 50-479, harboring the N88R mutation, was the first hIL-2 mutein tested clinically, which

resulted in the generation of ADAs[23]. Recently a PEGylated IL-2 molecule, termed NKTR-214, has been designed to circumvent the short half-life of IL-2 and provide selectivity toward the dimeric IL-2R. Although this molecule provides a kinetically controlled release of IL-2, PEGylation also interferes with CD122 and $\gamma_c$ binding, hampering the stimulation of effector cells[21]. Moreover, the large-scale manufacturing of such molecule can be difficult. To improve the pharmacokinetic properties of hIL-2 and minimize the stimulation of circulating immune cells, various immunocytokines have been developed that are made of wild-type or mutated hIL-2 coupled to targeting antibodies[17,19,20]. While linking to antibodies increases the half-life of hIL-2 muteins, this approach also enhances their risk of generating ADAs. Alternatively, immunocytokines can contain wild-type hIL-2, however, these do not preferentially stimulate CD122$^{high}$ effector lymphocytes and can engage with Treg and other immunosuppressive cells commonly found in the tumor microenvironment.

A different strategy from the aforementioned FPs and immunocytokines consists in the administration of hIL-2 in the form of IL-2cx. There, association of a very specific anti-IL-2 mAb with unmutated hIL-2 forms IL-2cx that result in IL-2R selectivity in vivo and reduced adverse side effects[8,10,11]. NARA1 is a recently-developed anti-hIL-2 mAb that precisely occupies the CD25-binding epitope of hIL-2; thus hIL-2/NARA1 complexes lead to preferential stimulation of CD122$^{high}$ effector T and NK cells[12], as well as expand dendritic cells[24]. Still, as with other CD122-biasing approaches, hIL-2/NARA1 complexes lead to detectable stimulation of Treg cells. The herein-presented NARA1leukin is a FP consisting of hIL-2 grafted onto the CDR1 of hNARA1's light chain.

When tested in vivo, NARA1leukin further increased stimulation of CD122$^+$ effector-type immune cells over what had been reported for hIL-2/NARA1 complexes while Treg cells remained unaffected. This correlated with improved anti-tumor responses, evident by the ratio of effector to regulatory cells in the tumor microenvironment. Interestingly, in comparison to hIL-2/NARA1 complexes, NARA1leukin was particularly efficacious in reducing metastatic tumor lesions. This finding is paramount when deciding on the preferred CD122-biased IL-2 candidate for clinical development and treatment of advanced cancer. One possible mechanism for NARA1leukin's pronounced property of reducing the burden metastatic lesions could rely on its exquisite CD122 bias and thus lack of stimulation of Treg cells. Several studies have demonstrated that Treg cells promote metastasis directly by interacting with cancer cells or creating an

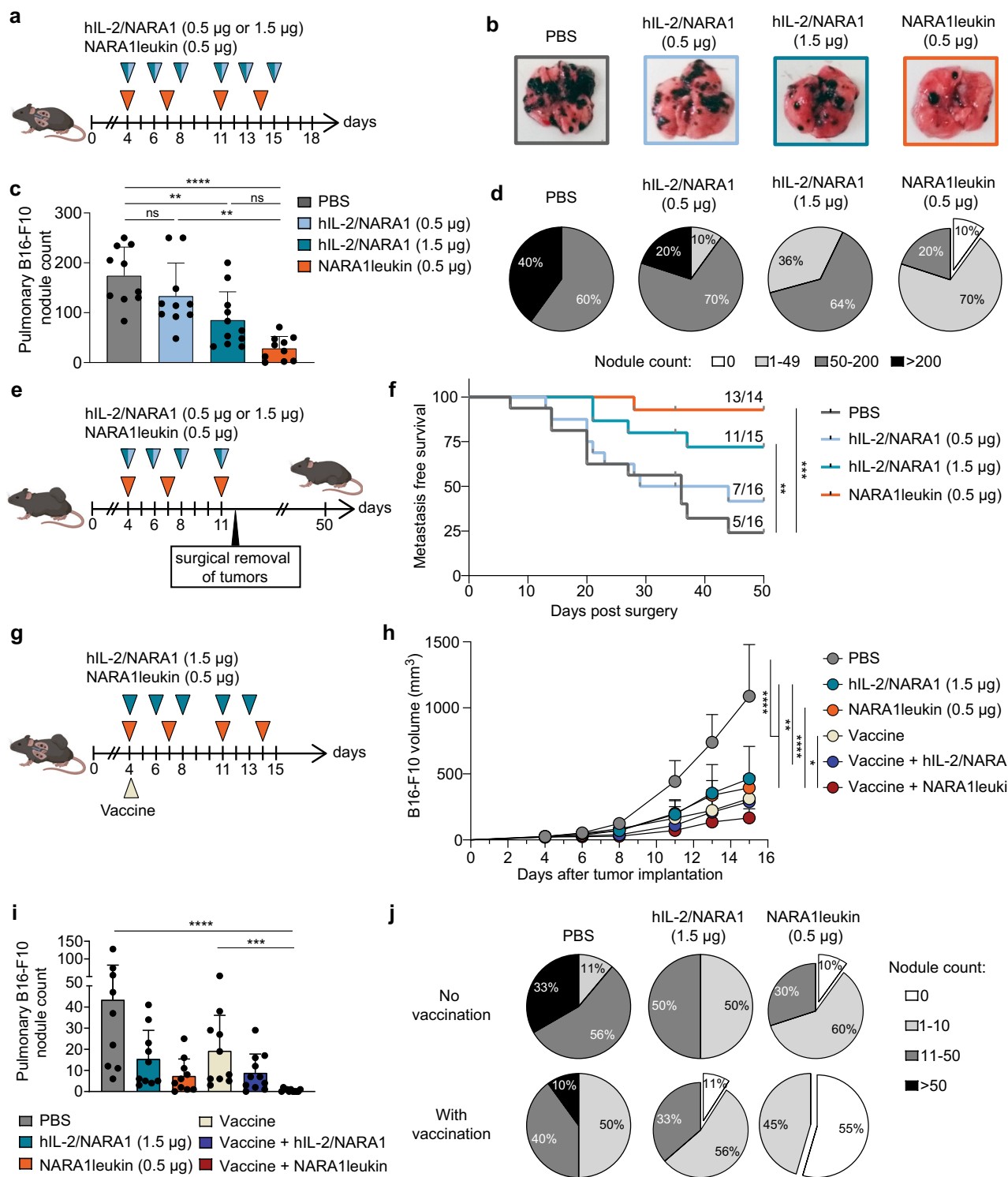

immunosuppressive environment in the primary or distant tumor site[25–29].

The results on NARA1leukin also provide insight into the mechanism of action of hIL-2/NARA1 complexes by demonstrating hIL-2 does not need to dissociate from NARA1 to bind and signal via CD122 and γ$_c$. While the design of NARA1leukin offers clear advantages over hIL-2/NARA1 complexes, such as irreversible blocking of IL-2's CD25-binding site and extended in vivo half-life, the short linker sequences introduced and structural modifications generated during engraftment of hIL-2 onto NARA1 can serve as neoepitopes that could trigger the

development of ADAs. The question of immunogenicity can only be addressed in clinical trials and should be noted as a possible limitation.

Combining NARA1leukin with other anti-cancer therapies offers attractive options. In this study we report synergistic effects of NARA1leukin in combination with peptide vaccination, which improved the control of both primary and metastatic tumors. Based on this and previous reports on CD122-biased IL-2 formulations, NARA1leukin could be tested together with (i) immune checkpoint blockers against cytotoxic T-lymphocyte-associated protein 4, PD-1 and TIM-3; (ii) vaccination; (iii)

**Fig. 5 Anti-tumor efficacy of NARA1leukin against metastatic disease. a** Treatment scheme. Mice were injected intravascularly with $3 \times 10^5$ B16-F10 melanoma cells and treated with PBS, hIL-2/NARA1 complexes (0.5 µg/2.5 µg or 1.5 µg/15 µg, three times weekly), or NARA1leukin (0.5 µg hIL-2 equivalent, twice weekly) from day 4 to 15. **b–d** Representative images of the lungs (**b**), quantification of lung nodules (**c**), and percentage distributions according to severity of metastasis categorized by nodule counts (**d**) are shown for day 18. ****$p < 0.0001$, **$p = 0.0456$ (PBS), **$p = 0.0024$ (hIL-2/NARA1). **e** Experimental scheme. Mice were injected with $10^6$ B16-F10 melanoma cells intradermally and treated as described in (**a**). On day 12 the intradermal tumor was surgically removed, and mice were observed for 35–50 days for lymph node metastasis. **f** Metastasis-free survival curves after surgery. Numbers indicate surviving mice out of total mice at endpoint. ***$p = 0.0005$, **$p = 0.0041$. **g** Experimental scheme. Mice were injected both intravascularly and intradermally with $3 \times 10^5$ and $10^6$ B16-F10 melanoma cells, respectively. Mice were treated with PBS, hIL-2/NARA1 complexes (1.5 µg /15 µg, three times weekly), or NARA1leukin (0.5 µg hIL-2 equivalent, twice weekly) from day 4 to 14 with or without peptide vaccination given on day 4. **h** Tumor growth curves of intradermal B16-F10 melanoma. ****$p < 0.0001$, **$p = 0.0087$, *$p = 0.0457$. **i, j** Quantification of lung nodules (**i**) and percentage distribution according to severity of metastasis categorized by nodule counts (**j**) are shown for day 15. ****$p < 0.0001$, ***$p = 0.0006$. Data are presented as mean ± SD of three independent experiments, $n = 10$ (PBS, hIL-2/NARA1 0.5 µg and NARA1leukin), $n = 12$ (hIL-2/NARA1 1.5 µg) (**b–d**), $n = 16$ (PBS and hIL-2/NARA1 0.5 µg), $n = 15$ (hIL-2/NARA1 1.5 µg), $n = 14$ (NARA1leukin) (**f**), $n = 9$ (PBS), $n = 10$ (hIL-2/NARA1, NARA1leukin, and Vaccine), $n = 11$ (Vaccine + hIL-2/NARA1 and Vaccine + NARA1leukin) (**h–j**) mice/group. Differences in nodule counts between groups were analyzed using Kruskal–Wallis test followed by Dunn's multiple comparison test (**c, i**). Differences in survival curves were analyzed by pairwise Mantel-Cox test (**f**). Differences between groups at the same time point were analyzed using two-way ANOVA followed by Tukey's multiple comparison test (**h**). ns, not significant. Source data are provided as a Source Data file.

radiotherapy; and (iv) epigenetic modifying agents[1,12,30–34]. Also, the properties of NARA1leukin are suitable for the development of bispecific antibodies for targeting tumor antigens and provision of CD122-biased IL-2 signals. Furthermore, the strategy used to generate NARA1leukin might be applicable to cytokines other than IL-2, where cytokine/anti-cytokine antibody complexes have also been reported to exert potent in vivo activity on mouse and human leukocyte subsets, including IL-1β, IL-3, IL-4, IL-6, IL-7, IL-15, and granulocyte colony-stimulating factor[35–43].

## Methods

**hIL-2, NARA1, and hIL-2/NARA1 complexes**. Human IL-2 (hIL-2; Proleukin) was purchased from R&D Systems or obtained from the National Cancer Institute's Biological Research Branch. hIL-2/NARA1 complexes were generated as previously described[12] and used at 2:1 or 1:1 molar ratios, as indicated. In humanized NARA1 (hNARA1), the variable regions of NARA1 were humanized as previously described[12]. We removed critical posttranslational modifications in the CDRs, designed several sequence variants, and cloned the resulting variable domains of heavy and light chains into a cytomegalovirus promoter-driven expression plasmid containing the constant regions of human wild-type IgG1, which were expressed in human embryonic kidney (HEK) 293 T cells. Sandwich ELISA for hIL-2 binding was used to select the best combination.

**Generation of NARA1leukin**. hIL-2 was split between K76 and N77 and was grafted into the CDR1 of the light chain of hNARA1 between Y31 and D34. Short gylcine linkers of 3⁻4 amino acid length were used to link the cytokine to the antibody. N-terminal half and C-terminal half of hIL-2 was linked to create a continuous construct. The resulting light chain was synthesized externally and expressed in HEK293-6E cells together with the parental heavy chain of NARA1leukin. Purification of NARA1leukin was done by affinity chromatography on Protein A sepharose followed by SEC.

**Cell cultures**. For protein expression in CHO-S cells, Power CHO-2, ProCHO-4 media and Ultraglutamine (Lonza) were used. Human PBMCs were isolated from Buffy coats using standard Ficoll density gradient isolation, and maintained in complete media with 10% fetal bovine serum (FBS), 1% L-glutamine, penicillin-streptomycin and 1% sodium pyruvate. Murine immune cells were isolated from C57BL/6 mouse spleens. B16-F10 melanoma, LLC, and CHO-S cell lines were purchased (ATCC and Thermo Scientific) and cultured in medium, consisting of advanced DMEM or RPMI 1640 with 10% FBS, 1% L-glutamine, penicillin-streptomycin, and Fungizone (all from Life Technologies).

**Enzyme-linked immunosorbent assay (ELISA)**. Flat Nunc-Immuno 96-well plates (Sigma-Aldrich) were coated overnight at 4 °C with capture antibodies in PBS: purified anti-hIL-2 mAb (clone 5344.111, BD Biosciences) or hCD25 (223-2 A/CF, R&D Systems). The plates were then washed with 0.1% Tween 20 (Sigma-Aldrich) and blocked for 1 hour at room temperature (RT) with 1% BSA (Sigma-Aldrich) in 0.1% Tween 20 at 450 rpm. After washing, titrated doses of hIL-2, hIL-2/NARA1 (hIL-2 and NARA1 were pre-mixed and incubated for 20 min) or NARA1leukin were added to plates in blocking solution and incubated for 1 to 2 h at RT and 450 rpm, followed by washing. For detection, purified biotinylated anti-hIL-2 mAb (clones MAB202, R&D System) in blocking solution was incubated for

1 h at RT and 450 rpm. Biotinylation was done using the EZ-link Sulfo-NHS.LC biotinylation kit (Thermo Scientific). Plates were then washed and incubated with streptavidin-conjugated horseradish peroxidase (BD Biosciences) for 45 minutes at RT in the dark. Following a final wash, plates were developed using TMB Peroxidase EIA substrate for 5–10 minutes (BioRad) until the reaction was stopped by the addition of 1.8 M $H_2SO_4$ (Sigma-Aldrich). The plates were read at 450 nm absorbance using an iMark microplate reader (BioRad).

**Protein characterization**. Superdex S200 column was used to perform analytical SEC of purified samples. In total 2–3 µg of the purified proteins were analyzed by SDS-PAGE using 4–20% NuPAGE Bis-Tris gels (BioRad) in reducing and non-reducing conditions following manufacturer's instructions with precision plus protein dual-color standard (BioRad).

**Human Samples**. Anonymized buffy coats were obtained from the Swiss Blood Bank Zurich. The "Fundamental research project for phenotypical and functional characterization of different leukocyte subsets in healthy and diseased individuals" (PFCL-1, BASEC no. 2016-01440) project has been reviewed and approved by the Kantonale Ethikkommission Zurich and has been carried out in accordance with principles enunciated in the current version of the Declaration of Helsinki, the guidelines of Good Clinical Practice, and Swiss legal requirements.

**STAT5 phosphorylation**. Mouse splenocytes or human PBMCs ($10^6$ cells/well) were seeded in 96-well plates and stimulated for 15 min at 37 °C, using hIL-2, hIL-2/NARA1 complex, or NARA1leukin at equimolar hIL-2 amounts. Cells were directly fixed with Fix Buffer I (BD) for 10 min at 37 °C followed by permeabilization with Perm Buffer III (BD Phosflow) for 1 h and stained with fluorochrome-conjugated antibodies in flow cytometry buffer.

**Flow cytometry**. Single cell suspensions of spleens or lymph nodes (LNs) were prepared according to standard protocols. Tumors were processed as previously described[12]. Briefly, tumors were cut into small pieces, and incubated in 10 ml dissociation buffer (RPMI, 5% FCS, 10 ug/ml DNAse I (Sigma-Aldrich), and 200 U/ml collagenase type I (Thermo Fisher Scientific) for 60 min at 37 °C and shaking with 25 rpm. Cell suspensions were then passed through a 70 µm cell strainer, after one wash a Percoll (40% and 70%; GE Healthcare) gradient centrifugation was performed. Lungs were passed through a 70 µm cell strainer followed by Percoll gradient centrifugation. Total cell counts in organs were determined by using a BioRad TC20 Automated Cell Counter. Dead cells were excluded using Fixable Viability Dye eFluor™ 780 (eBioscience™). Intracellular, Foxp3, and Ki67 staining was performed following manufacturer's instructions. For in vivo pSTAT5 detection, spleens were directly fixed in Lyse/Fix Buffer (BD Phosflow™) followed by permeabilization with Perm Buffer III (BD Phosflow) for 1 h. Samples were acquired on a BD LSR II flow cytometer (BD Biosciences). The data are represented as percentages of the parent gate or as total cell counts calculated based on frequencies multiplied by the total organ cell counts.

**Mice**. 2-3-month old female C57BL/6 mice were purchased from Charles River Laboratories. Balb/c mice were bred in-house and used for experiment at 2-3-months of age. All mice were kept in specific pathogen-free (SPF) conditions at the Biologisches Zentrallabor (BZL) where they were maintained at 22 °C and on a 12-hour light-dark cycle (7 am to 7 pm) and given food and water ad libitum. Experiments followed the Swiss Federal Veterinary Office guidelines and were

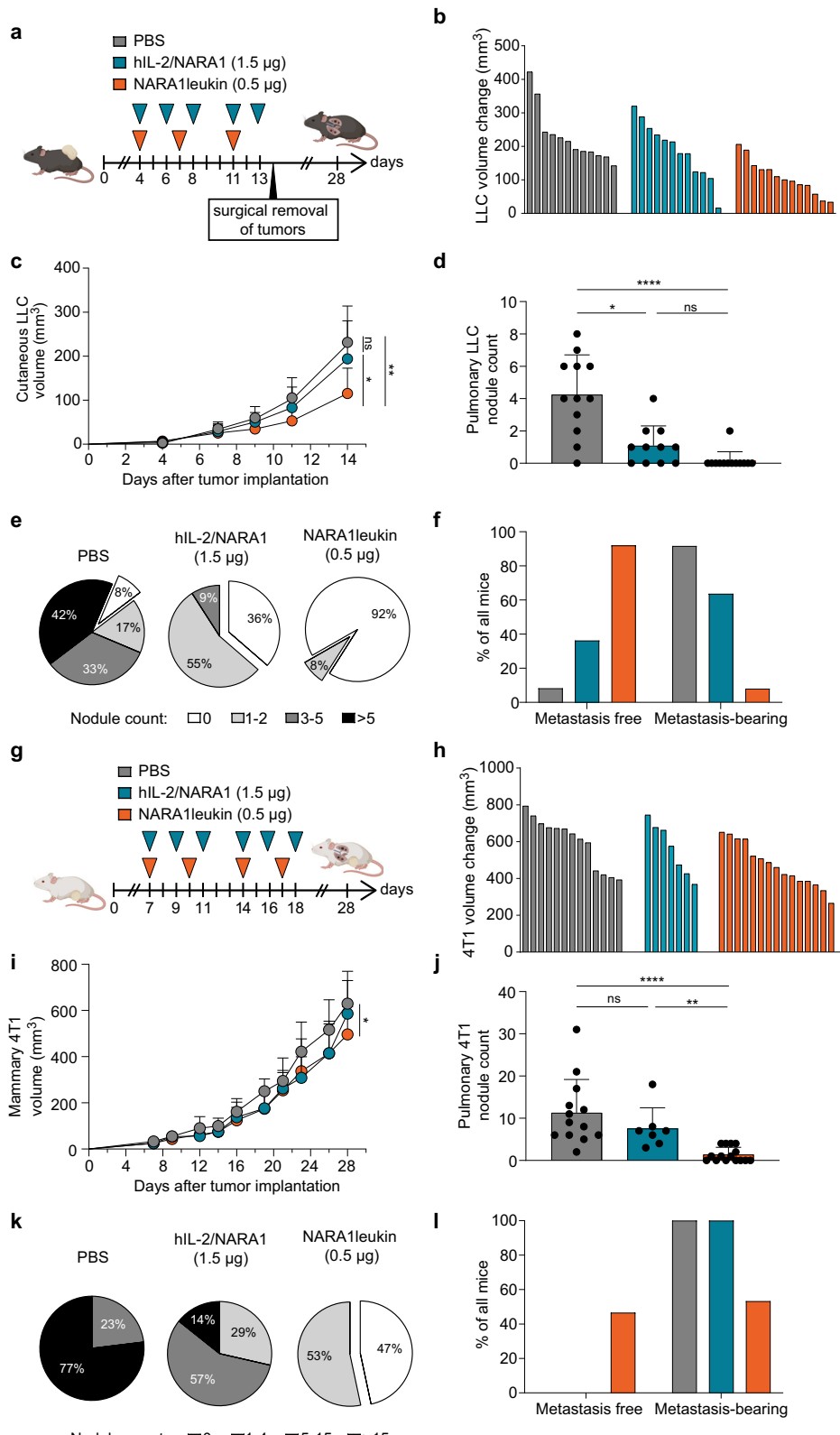

approved by the Cantonal Veterinary Office. Pre-established exclusion criteria followed these offices guidelines.

**In vivo treatments**. All cytokine treatments were injected intraperitoneally. Doses and schedules used for every experiment are indicated in the figure legends. The indicated doses correspond to the amount of hIL-2 in free hIL-2, a complex or a fusion protein. Briefly, the treatments are phosphate-buffered saline (PBS), hIL-2 (1.5 μg, corresponding to 100 μmol), hIL-2/NARA1 complexes as 2:1 (0.5 μg hIL-2

mixed with 2.5μg NARA1 in PBS, corresponding to 33 μmol hIL-2 and 16.7 μmol NARA1, or 1 μg hIL-2 mixed with 5 μg NARA1, corresponding to 66 μmol hIL-2 and 33 μmol NARA1) or 1:1 molar ratios (1.5 μg hIL-2 with 15 μg NARA1 in PBS, corresponding to 100 μmol hIL-2 and 100 μmol NARA1) or NARA1leukin (0.5 or 1 μg hIL-2 equivalent, corresponding to 16.7 μmol or 33 μmol NARA1leukin). The peptide vaccination composed of 70 nmol gp100-peptide (AVGA-LEGPRNQDWLGVPRQL), 100 μg anti-mouse CD40 antibody (FGK45, BioXCell) and 100 μg Poly I:C (Sigma Aldrich) and injected subcutaneously into the flank.

**Fig. 6 Efficacy of NARA1leukin against spontaneous metastasis.** (**a**) Experimental scheme. Mice were intradermally injected with $10^6$ LLC cells and treated with PBS, hIL-2/NARA1 complexes (1.5 μg/15 μg, three times weekly), or NARA1leukin (0.5 μg hIL-2 equivalent, twice weekly) from day 4 to 13. On day 14 the intradermal tumor was surgically removed, and mice were observed for 28 days. (**b, c**) Change in tumor volume in individual mice between day 4 and 14 (**b**) and tumor growth curves until day 14 (**c**) are shown. **$p = 0.0018$, *$p = 0.0392$. (**d, e**) Quantification of spontaneous lung nodules on day 28 (**d**), percentage distribution according to severity of metastasis categorized by nodule counts (**e**) are shown. ****$p < 0.0001$, *$p = 0.0325$. (**f**) Percentages of metastasis-free and metastasis-bearing mice at endpoint. (**g**) Experimental scheme. Mice were injected with 4T1 cells within mammary fat pad and treated as described in (**a**) until day 18. (**h, i**) Change in tumor volume in individual mice between day 7 and 28 (**h**) and tumor growth curves until day 28 (**i**) are shown. *$p = 0.0338$. (**j, k**) Quantification of spontaneous lung nodules on day 28 (**j**), percentage distribution according to severity of metastasis categorized by nodule counts (**k**) are shown. ****$p < 0.0001$, **$p = 0.0079$. (**l**) Percentage of metastasis-free and metastasis-bearing mice at endpoint. Data are presented as mean ± SD of three independent experiments (two for hIL-2/NARA1 group of 4T1), $n = 12$ (PBS), $n = 11$ (hIL-2/NARA1), $n = 13$ (NARA1leukin) (**b–f**), $n = 13$ (PBS), $n = 7$ (hIL-2/NARA1), $n = 15$ (NARA1leukin) (**h–l**) mice/group. Differences between groups at the same time point were analyzed using mixed effects analysis with Greenhouse-Geisser correction followed by Tukey's multiple comparison test (**c, i**). Differences in nodule counts between groups were analyzed using Kruskal-Wallis test followed by Dunn's multiple comparison test (**d, j**). ns, not significant. Source data are provided as a Source Data file.

**Tumor models.** For the cutaneous melanoma model, $10^6$ B16-F10 melanoma cells were injected intradermally into the back of syngeneic C57BL/6 mice. For pulmonary nodules, $3 \times 10^5$ B16-F10 melanoma cells were injected intravascular through the tail vein to syngeneic C57BL/6 mice. $3 \times 10^5$ LLC cells were injected intradermally into the back of syngeneic C57BL/6 mice. Primary LLC tumors were removed surgically after 15 days to observe spontaneous lung metastasis on day 28. $10^5$ 4T1 mammary cancer cells were injected to the mammary fat pad of syngeneic Balb/c mice. Tumor volume was calculated as follows: V = 2/3 × π × ((a + b)/4)[3], with "a" indicating length and "b" width of tumor (both in mm). Treatment of mice was started when tumors became visible and palpable (on day 4 for B16-F10 and LLC, on day 7 for 4T1) and consisted of 4-6 injections 2-3 times per week of PBS, hIL-2/NARA1 or NARA1leukin, as indicated. Lungs were stained with Bouin's solution overnight to better visualize LLC and 4T1 nodules.

**Statistical analyses.** Mice were randomly assigned to experimental groups and results are shown as mean ± SD. Statistical analyses were performed in GraphPad Prism 8. The details of performed statistical tests are provided in the figure legends. Comparison of multiple groups with unmatched data were done by Kruskal-Wallis test followed by Dunn's multiple comparison test. Comparisons of tumor growth curves with matched measurements were done either by mixed effects analysis with Greenhouse-Geisser correction or two-way ANOVA followed by Tukey's multiple comparison test.

**Software.** Flow cytometry data has been collected with BD FACSDiva™ 8.0.1. Software and analyzed using BD FlowJo 10 and Microsoft Excel 16. Data were plotted and statistically analyzed using GraphPad Prism 8. Display items were created using Biorender.com.

**Material availability.** Materials that are not available commercially can be requested from the corresponding author.

**Reporting summary.** Further information on research design is available in the Nature Research Reporting Summary linked to this article.

## Data availability
Data are available within the Article, Supplementary Information or available from the authors upon request. Source data are provided with this paper.

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

## Acknowledgements

We thank Laura Bürgi for help with experiments, Ulrike Held and Lukas Heeb for help with statistical analyses, and Miro Raeber for preparing the ethical approval for human samples and display items. We further acknowledge helpful discussions with Barbara Brannetti, Thomas Calzascia, Andreas Katopodis, Jiri Kovarik, Thomas Pietzonka, Simone Popp, Catherine Regnier, Jean-Michel Rondeau, Emmanuelle Wirth, Gerhard Zenke, and Chao Zou (all from Novartis, Basel) who contributed to a previous publication[12] on NARA1 and a patent application[13] describing IL-2–NARA1 FPs, including NARA1leukin. This work was funded by Swiss National Science Foundation grant 310030-172978 (to O.B.), Swiss Cancer Research grants KFS-4136-02-2017 and KFS-5028-02-2020 (both to O.B.), Hochspezialisierte Medizin Schwerpunkt Immunologie (HSM-2-Immunologie; to O.B.), Helmut Horten Foundation (to O.B.), Cancer Research Foundation (to N.A.R.), and Sassella Foundation (to N.A.R.).

## Author contributions

D.S. (Figs. 1–6), N.A.R. (Figs. 1–3), M.R. (Figs. 2–4), U.K. (Figs. 2, 3), M.H. (Fig. 2), M.G. (Fig. 6), and L.B. (Fig. 6) designed, performed, and analyzed experiments. O.B. conceived the project, designed, and analyzed experiments. D.S. and O.B. wrote the final manuscript, with contributions of all other authors.

## Competing interests

O.B. is a founder and shareholder of Anaveon AG. The remaining authors declare no competing interests.
