## [Peer Review File · Nature Communications]

REVIEWERS' COMMENTS

Reviewer #1 (Remarks to the Author):

Please comment on the relative molar concentrations of each therapy given as direct ug dosages do not consider a large disparity between the molecular weight of hIL-2 and IL-2 immunocomplexes/fusion proteins. Please include the method by which the authors quantify lymphocyte counts.

Reviewer #3 (Remarks to the Author):

The pSTAT5 dose response curves in Fig. 1 and Fig.5E were statistically analyzed by comparing the data at each point in the curve; these results were shown in Fig. E1 and E5. These types of dose-response data are better analyzed by performing non-linear regression of the resulting curves and calculating the EC50s for the responses by each cell population to the various forms of IL-2. This would result in a single value for each curve and would better define the responses. From the EC50s then statistical analysis can be performed to determine whether the responses significantly varied.

Reviewer #1:

Please comment on the relative molar concentrations of each therapy given as direct ug dosages do not consider a large disparity between the molecular weight of hIL-2 and IL-2 immunocomplexes/fusion proteins.

Response: We thank the reviewer for pointing out this aspect. The doses that are indicated as “ug” refer to the amount of hIL-2 in free hIL-2, a complex or a fusion protein, which therefore accounts for the different molecular weights of the constructs. We have now added the molar equivalents of the IL-2 treatments to our Methods section and clarified this issue also in the text on lines 133-135.

Please include the method by which the authors quantify lymphocyte counts.

Response: We thank the reviewer for this comment. We have now included this information in the Methods section.

Reviewer #3 (Remarks to the Author):

The pSTAT5 dose response curves in Fig. 1 and Fig.5E were statistically analyzed by comparing the data at each point in the curve; these results were shown in Fig. E1 and E5. These types of dose-response data are better analyzed by performing non-linear regression of the resulting curves and calculating the EC50s for the responses by each cell population to the various forms of IL-2. This would result in a single value for each curve and would better define the responses. From the EC50s then statistical analysis can be performed to determine whether the responses significantly varied.

Response: We thank the reviewer for this comment. We have now statistically re-analyzed the data and indicated the differences in curves/EC50s using one-way ANOVA.